# Knockout of *sws2a* and *sws2b* in Medaka (*Oryzias latipes*) Reveals Their Roles in Regulating Vision-Guided Behavior and Eye Development

**DOI:** 10.3390/ijms24108786

**Published:** 2023-05-15

**Authors:** Ke Lu, Jiaqi Wu, Shulin Tang, Xiaodan Jia, Xu-Fang Liang

**Affiliations:** 1College of Fisheries, Chinese Perch Research Center, Huazhong Agricultural University, Wuhan 430070, China; luke@webmail.hzau.edu.cn (K.L.);; 2Engineering Research Center of Green Development for Conventional Aquatic Biological Industry in the Yangtze River Economic Belt, Ministry of Education, Wuhan 430070, China

**Keywords:** *sws2a*, *sws2b*, behavior, eye development, medaka

## Abstract

The medaka (*Oryzias latipes*) is an excellent vertebrate model for studying the development of the retina. Its genome database is complete, and the number of opsin genes is relatively small compared to zebrafish. Short wavelength sensitive 2 (*sws2*), a G-protein-coupled receptor expressed in the retina, has been lost in mammals, but its role in eye development in fish is still poorly understood. In this study, we established a *sws2a* and *sws2b* knockout medaka model by CRISPR/Cas9 technology. We discovered that medaka *sws2a* and *sws2b* are mainly expressed in the eyes and may be regulated by growth differentiation factor 6a (*gdf6a*). Compared with the WT, *sws2a*^−/−^ and *sws2b*^−/−^ mutant larvae displayed an increase in swimming speed during the changes from light to dark. We also observed that *sws2a*^−/−^ and *sws2b*^−/−^ larvae both swam faster than WT in the first 10 s of the 2 min light period. The enhanced vision-guided behavior in *sws2a*^−/−^ and *sws2b*^−/−^ medaka larvae may be related to the upregulation of phototransduction-related genes. Additionally, we also found that *sws2b* affects the expression of eye development genes, while *sws2a* is unaffected. Together, these findings indicate that *sws2a* and *sws2b* knockouts increase vision-guided behavior and phototransduction, but on the other hand, *sws2b* plays an important role in regulating eye development genes. This study provides data for further understanding of the role of *sws2a* and *sws2b* in medaka retina development.

## 1. Introduction

Visual pigments in photoreceptor cells are very important in light detection and signal transduction [1,2]. Visual pigments consist of opsin receptor proteins that are covalently bound to vitamin A-derived photochromophores, and opsins mediate light absorption in cones and rods [3]. In vertebrates, rod cells responsible for dark or low-light vision express only rhodopsin (*rh1*), while cone cells for bright or well-lit vision fall into four main categories: short- (*sws1*, *sws2*), medium- (*rh2*), and long- (*lws*) wavelength-sensitive opsins [4]. Several studies have shown that opsins play a crucial part in feeding, behavior, and color sensitivity in fish [5,6,7,8]. In the process of fish evolution, some copies of the opsin gene may be lost, but all opsins are basically present, while *sws2* and *rh2* are lost in most mammals [9,10]. Therefore, the function of *sws2* and *rh2* in fish is worth studying. *RH2* mainly performs its visual function in fish that are active at night or dusk [11]. Much less is known about *sws2*, with only Percomorpha fish having received extensive research [12].

Visual pigment proteins show great diversity under natural selection, and many fish genomes contain one to three copies of *sws2* genes [13,14,15]. In rainbow trout (*Oncorhynchus mykiss*), *sws2* is switched from *sws1*, a process that begins before the yolk sac is fully absorbed and continues throughout the juvenile period [16,17]. Atlantic Cod (*Gadus morhua*) have lost *sws1* and *lws* opsins in evolution, and only uses *sws2* and *rh2* opsins to detect prey, avoid predators, and adapt to light response [18]. Similarly, the threespine stickleback (*Gasterosteus aculeatus*) adapts its visual perception to blackwater habitats by tuning spectra at *sws2* sites [14]. There is a group of weakly electric teleost fishes in South America (Gymnotiforms) that are nocturnal, many living in muddy streams or deep rivers, and which have lost the *sws1* and *sws2* opsin genes through evolution [19]. In addition, changes in opsin expression patterns and gene expression regulation also affect opsin function.

The expression of opsin varies greatly among different fish species, but each species may change according to ontogeny and changes in the light environment [20]. On the other hand, the regulation of *sws2* gene expression is also controversial. Both *gdf6a* and forkhead box Q2 (*foxq2*) have been proven to regulate the expression and differentiation of *sws2* [21,22]. In addition, some studies have shown that changes in the opsin genes affected by transcriptional regulators or drug exposure can also cause changes in visual guidance behavior [23,24].

The medaka (*Oryzias latipes*) is an ideal model organism for studying visual development, and its retina is rich in various types of cone photoreceptors that mediate color perception [25,26]. Medaka and zebrafish (*Danio rerio*) retinas exhibit distinct patterns of photoreceptor cell mosaics, with each blue-sensitive cone (*sws2*) surrounded by four double cones (*lws* and *rh2*) [27]. Despite the complexity of the spatiotemporal pattern of cone opsin expression in fish, recent genome-editing technologies have made the use of mutants in genome studies possible [28,29].

In this study, we generated *sws2a* and *sws2b* knockout medaka models using CRISPR/Cas9. We discovered that medaka *sws2a* and *sws2b* are mainly expressed in the eyes and may be regulated by *gdf6a*. The enhanced vision-guided behavior in *sws2a*^−/−^ and *sws2b*^−/−^ medaka larvae may be related to the enhanced photoconductive signaling. Additionally, we also found that *sws2b* affects the expression of eye development genes, while *sws2a* does not. These findings indicate that *sws2a* and *sws2b* knockouts increase vision-guided behavior and upregulated phototransduction-related genes, but on the other hand, *sws2b* plays an important role in regulating eye development genes.

## 2. Results

### 2.1. Expression of sws2a and sws2b in Medaka

There are two orthologous genes of *sws2* in medaka, namely *sws2a* and *sws2b,* and sequence alignment showed the amino acid residues of medaka *sws2a* and *sws2b* had 77.27% homology (Figure 1A). In addition, the expression of *sws2a* and *sws2b* mRNA in different tissues was detected by RT-PCR. Both *sws2a* and *sws2b* were expressed mainly in the eyes among the adult tissues (Figure 1B). The expression levels of *sws2a* and *sws2b* were low in larval fish, while *sws2b* expression was increased in adult fish (Figure 1C).

### 2.2. Establishment of sws2a and sws2b Mutant Medaka

To understand the importance of *sws2a* and *sws2b* in visual performance, we generated *sws2a* and *sws2b* mutant medaka by CRISPR/Cas9 technology. The CRISPR/Cas9 guide-RNAs close to the translation start codon were selected to change the base sequences of *sws2a* and *sws2b*, thereby blocking protein translation. Compared with the WT medaka, *sws2a* and *sws2b* homozygous mutants showed 4 bp and 274 bp deletions, respectively, and the deletions both resulted in the early termination of translation of the entire seven-transmembrane domain (7tm_1) in SWS2 (Figure 2A–D). Protein structure prediction further elucidated the knockout results (Figure 2E,F).

In order to determine whether the *sws2a*^−/−^ and *sws2b*^−/−^ mRNA was decayed, we analyzed the transcript levels of *sws2a* and *sws2b* in WT, *sws2a*^−/−^, and *sws2b*^−/−^ medaka by RT-qPCR. The *sws2a*^−/−^ and *sws2b*^−/−^ medaka both had no significant decrease in *sws2a* and *sws2b* mRNA levels, respectively, while transcriptional compensation of *sws2b* deletion was observed *sws2a*^−/−^ (Figure 3A,B). Moreover, we designed two pairs of total-length primers to distinguish the expression differences between the two transcripts of medaka *sws2* (Appendix A). Next, we sequenced the total lengths of *sws2a* and *sws2b* in *sws2a*^−/−^ and *sws2b*^−/−^ medaka, which showed 4 bp and 274 bp deletions in *sws2a* and *sws2b* mRNA levels in *sws2a*^−/−^ and *sws2b*^−/−^ mutants (Appendix A). *sws2a*^−/−^ and *sws2b*^−/−^ showed no morphological difference from the WT medaka (Appendix A).

Because *gdf6a* and *foxq2* were reported to determine the expression of *sws2* [21,22], we analyzed the mRNA expression levels of *gdf6a* and *foxq2* in *sws2a*^−/−^ and *sws2b*^−/−^ mutants. *GDF6A* mRNA expression levels were significantly increased in both *sws2a*^−/−^ and *sws2b*^−/−^ compared to WT (Figure 3C, *p* < 0.05). However, the mRNA expression levels of *foxq2* exhibited no significant difference in the fish (Figure 3D, *p* > 0.05).

### 2.3. Feeding and the Behavioral Tests

The food intake of *Artemia* in *sws2a*^−/−^ medaka larvae was significantly higher than that of WT and *sws2b*^−/−^ (Figure 4A, *p* < 0.05), but there was no significant difference in growth (Figure 4B, *p* > 0.05).

We next examined the visual responsiveness in *sws2a*^−/−^ and *sws2b*^−/−^ mutants at 6 dph. Compared with the WT, *sws2a*^−/−^ and *sws2b*^−/−^ mutant larvae displayed an increase in swimming speed during the changes from light to dark (Figure 4C, *p* < 0.05). We also observed that *sws2a*^−/−^ and *sws2b*^−/−^ larvae both swam faster than WT in the dark and the first 10 s of the 2 min light period (Figure 4D, *p* < 0.05). They all had a peak in the first 10 s of the 2 min light period, while the swimming speed was detected to increase by 48.1% and 57.3% in *sws2a*^−/−^ and *sws2b*^−/−^ larvae, respectively.

### 2.4. Transcript Levels of Phototransduction-Related Genes in Larvae of sws2a^−/−^ and sws2b^−/−^ Mutants

We examined the mRNA levels of phototransduction-related genes in larvae of the WT, *sws2a*^−/−^, and *sws2b*^−/−^ medaka by RT-qPCR. Surprisingly, the deletion of *sws2a* or *sws2b* did not affect the expression of other cone and rod genes (Figure 5A, *p* > 0.05). We further measured the mRNA levels of cone phototransduction genes in WT, *sws2a*^−/−^, and *sws2b*^−/−^ medaka at 6 dph. Expression of guanine nucleotide-binding protein (G protein), beta polypeptide 3b (*gnb3a*), and G protein-coupled receptor kinase 7a (*grk7a*) were increased significantly in the *sws2b*^−/−^ medaka, while only *grk7a* was elevated in *sws2a*^−/−^ larvae (Figure 5B, *p* < 0.05).

### 2.5. Transcript Levels of Eye Development Gene in Larval sws2a and sws2b Knockout Medaka

We next tested whether *sws2a* and *sws2b* knockout affected the expression of the eye development genes in the larvae. Compared to WT, *sws2a*^−/−^ larvae displayed an increased mRNA expression pattern of SIX homeobox 7 (*six7*) (Figure 6A, *p* < 0.05), while no alteration was observed in the *sws2b*^−/−^. In addition, the transcriptional expression levels of paired box 6 (*pax6*), SIX homeobox 3a (*six3a*), and *six3b* were significantly decreased in the *sws2b*^−/−^ larvae (Figure 6B–D, *p* < 0.05).

## 3. Discussion

In contrast to most mammals that have lost *sws2* and *rh2* genes over evolutionary time, all four opsin gene types are present in whole-genome duplication in teleost fish, although there is a phenomenon of gene copy loss [30,31]. *SWS2A* and *sws2b* were developed by the tandem duplication of *sws2* in the common ancestor of Neoteleostei fishes [32]. For instance, despite having highly conserved *foxl2a* and *foxl2b* domains in zebrafish, these two genes have divergent functions and synergies. Disruption of *foxl2a* and *foxl2b* led to premature ovarian failure and partial sex reversal, respectively, and *foxl2a* and *foxl2b* jointly regulate the development and maintenance of zebrafish ovaries [33]. The structural domain of *sws2a* and *sws2b* proteins in medaka was also highly conserved, and the protein similarity between the two proteins reached 77.27%. Medaka *sws2a* and *sws2b* are mainly expressed in the eye during adulthood, while *sws2b* expression is significantly elevated in adult fish relative to larva. This is consistent with the expression pattern of *sws2a* and *sws2b* in bluefin killifish (*Lucania goodei*) [34].

In this study, we produced and characterized *sws2a* and *sws2b* medaka mutant lines with 4 bp and 274 bp deletions that caused the loss of seven transmembrane domains in SWS2A and SWS2B. We detected that the transcription levels of *sws2a* and *sws2b* were not affected by the mutation. Although we did not find specific antibodies for medaka SWS2A or SWS2B that confirm them at the protein level, we further amplified the total lengths of the gene using cDNA as a template, and sequenced it. The results showed that *sws2a* and *sws2b* mutants had 4 bp and 274 bp deletions of *sws2a* and *sws2b* mRNA, respectively. These results indicated that the knockouts of *sws2a* and *sws2b* in this study are effective. Our results are in accordance with the recent work on *lca5* knockout zebrafish [35]. *SWS2* gene and protein expressions are regulated by upstream signals [36]. *GDF6A* is a member of the bone morphogenetic protein family that induces dorsal retinal differentiation during ocular morphogenesis [37]. *GDF6A* deletion zebrafish larvae displayed fewer blue cone photoreceptor cells, and *gdf6a* could develop and maintain the *sws2* [21]. Disruption of *foxq2* in zebrafish showed the loss of *sws2* cone expression in 5-day post-fertilization (dpf) larvae, and further results indicated that *foxq2* was an activator of *sws2* transcription and inhibited *sws1* expression [22]. In addition, the thyroid hormone (TH) may induce the transition of *sws1* to *sws2* opsin by binding to its receptor *thrβ2* [38]. Growth hormone (GH) can accelerate the process of *sws1* to *sws2* opsin conversion [39]. Here, we observed that *gdf6a* expression was significantly upregulated in *sws2a*^−/−^ and *sws2b*^−/−^ mutants, while *foxq2* expression was unaffected. We speculated that *sws2a* and *sws2b* are mainly regulated by *gdf6a* in medaka, mainly because when *sws2a* or *sws2b* was knocked out, its upstream gene *gdf6a* expression increased through negative feedback regulation. Similar to *foxl2* and *sox9*, *foxl2* is downstream of *sox9*, and *sox9* transcript level significantly increased in *foxl2a*^−/−^ and *foxl2b*^−/−^ zebrafish mutants [33,40].

Feeding assessment showed increased food intake in *sws2a*^−/−^ mutant zebrafish, while *sws2b*^−/−^ was unaffected. In a previous report, *tbx2b* zebrafish mutants (reduced UV (SWS1) cones) showed declined foraging performance [41]. Another study demonstrated that *sws2* and *rh2* opsins decreased, and visual prey capture was impaired in *six6a*/*six6b*/*six7* triple-knockout zebrafish at 6 dpf [23]. It has also been proposed that in the case of *sws1* regulation, the increase in *sws2* may be related to prey search [5,42]. Therefore, we think that the increased intake of *sws2a^−/−^* may be related to the deletion of *sws2a*. Interestingly, the feeding rate and survival rate of larvae of haddock (*Mellanogrammus aeglefinus*) increased under blue and green light compared with other light colors [43,44]. Light/dark motion tests and responses to light stimuli are effective methods to reflect the integrated function of visual pathways [45,46,47]. In this study, compared with the control group, *sws2a*^−/−^ and *sws2b*^−/−^ mutants both significantly increased swimming speed when they changed from light to darkness. When the larvae were suddenly stimulated by light, *sws2a*^−/−^ and *sws2b*^−/−^ mutant larvae both reacted strongly within the first 10 s. The swimming speed increased by 48.1% and 57.3% in *sws2a*^−/−^ and *sws2b*^−/−^ larvae, respectively. These results are consistent with our finding that cone phototransduction pathway-related genes are further activated in *sws2a*^−/−^ and *sws2b*^−/−^ larvae. *GRK7A* is mainly expressed in the outer cone segment; *grk7* knockdown in larval zebrafish caused a delay in dark adaptation and impaired cone response recovery [48]. Moreover, *grk7a* has been shown to be involved in the recovery of cone light response in larval zebrafish [49]. Additionally, GNB3 is a G-protein beta subunit located in the outer segment of cones and mainly plays a role in the phototransduction cascade of cones [50]. We concluded that the enhanced vision-guided behavior in *sws2a*^−/−^ and *sws2b*^−/−^ medaka larvae may be the result of upregulated phototransduction genes.

In vertebrates, opsins are G-protein-coupled receptors expressed in the retina, responsible for promoting eye sensitivity to light. Each cone opsin covers a distinct part of the visible spectrum, with corresponding non-absorption maxima [51]. Our RT-PCR analysis indicated that the loss of *sws2a* and *sws2b* did not affect the normal expression of other opsins. In addition to being absorbed by *sws2*, blue light may also be absorbed by neighboring *sws1* and *rh2*. Thus, the sensitivity of blue light is not controlled by *sws2* [52]. Fish such as the southern catfish (*silurus meridionalis*), which are mainly nocturnal and live in underground or even burrowing freshwater environments, lost *sws2* [53,54]. Some recently studied cartilaginous fishes also show no expression of *sws2*, further supporting the hypothesis *that sws2* was lost early in the cartilaginous lineage [55,56,57]. However, *sws2* plays an irreplaceable role in some species [14]. The Japanese flounder (*Paralichthys olivaceus*) adjusted their *sws2a* and *rh2* genes as their light environment changed during development [58]. Therefore, diverse visual systems are adaptive responses to varying environments in different species.

In this study, there was no significant retinal histological damage observed during the retinal histological examination, indicating that its effect may be at the molecular level. *SIX3*, *six6,* and *six7* are important regulators of fish retinal development and differentiation [23,59]. Knock-down *six3b* and *six7* resulted in the loss of eyes, whereas disruption of *six3a* and *six7* had no significant eye phenotypes [60]. A previous study demonstrated that *six6* deletion mice showed severe retinal abnormalities [61]. *PAX6* is necessary for the normal development of fish eyes [62]. Deletion of a single copy of *pax6* in mice showed microphthalmia, while mutation with a double copy showed anophthalmia [63]. In the present study, we found that *six3a*, *six3b,* and *pax6b* were downregulated in *sws2b*^−/−^ medaka larvae, suggesting that the loss of *sws2b* may affect the retinal development of medaka larvae. In addition, the upregulated expression of *six7* in *sws2a*^−/−^ medaka larvae might be due to the regulation of *sws2a* by *six7* [64]. Paradoxically, *sws2b*^−/−^ medaka larvae showed increased motor capacity, but decreased transcription levels of eye development and regulatory genes. Our explanation is that *sws2b* may delay the movement of medaka larvae, but on the other hand, *sws2b* plays an important role in regulating eye development genes. The mechanism of medaka *sws2a* and *sws2b* double knockout in regulating eye development needs to be determined in future studies.

In summary, this study established *sws2a* and *sws2b* knockout medaka by CRISPR/Cas9 technology. We speculated that the enhanced vision sensitivity in regulating vision-guided behavior in *sws2a* and *sws2b* knockout medaka larvae might be related to the upregulation of phototransduction-related genes. Additionally, *sws2b* affects eye development gene expression, implying different mechanisms of *sws2a* and *sws2b*. This study provides data for further understanding of the role of *sws2a* and *sws2b* in medaka retina development.

## 4. Materials and Methods

### 4.1. Medaka Lines and Maintenance

The wild-type (WT) medaka are an orange strain, and were maintained in an environment of 26~28 °C and 14 light/10 h dark cycle. Medaka embryos were cultured at 27~28 °C in medaka embryo medium (MEM) [65]. The 6 dph (day post-hatching) larvae were fed with live *Artemia* twice daily when the yolk sac was almost completely consumed. All the fish were anesthetized with tricaine methanesulfonate (MS-222) before the tissue collection. 

### 4.2. Generating sws2a^−/−^ and sws2b^−/−^ Mutants by CRISPR/Cas9 Technology

Medaka *sws2a* (ENSORLG00000040205) and *sws2b* (ENSORLG00000028370) genes were targeted using CRISPR/Cas9 technology. The sequencing of single-guide RNAs (sgRNAs) and PCR primers are shown in Appendix A. sgRNAs were cloned into pMD-18T vector and were synthesized using TranscriptAid T7 High Yield Transcription kit (ThermoFisher Scientific, Waltham, MA, USA). The compounds of sgRNAs (50 ng/µL) and Cas9 protein (New England Biolabs, Ipswich, MA, USA) were co-injected into one- or two-cell stage wild-type embryos. The F0 medaka were raised to adulthood and outcrossed with wild-type to produce F1 medaka. A T7 endonuclease 1 assay (Vazyme, Nanjing, China) was used to detect *sws2a* heterozygous mutant individuals according to the manufacturer’s instructions. The heterozygous individuals of *sws2b* mutation with large fragment deletion were distinguished by PCR detection and then sequenced. The F1 heterozygous individuals in-crossed to generate F2 homozygous individuals, and all experiments were conducted with F3 homozygous individuals. Unless otherwise mentioned, the homozygous mutant lines of *sws2a* and *sws2b* in subsequent experiments were addressed as *sws2a*^−/−^ and *sws2b*^−/−^, respectively. The SWISS_MODEL (https://swissmodel.expasy.org/interactive) (1 March 2023) was used to predict the protein tertiary structures. The protein tertiary structures of *sws2a* and *sws2b* were predicted using the SWISS_MODEL (https://swissmodel.expasy.org/).

### 4.3. Larvae Feeding Assays

For larvae food intake, 6 dph larvae were fed with *Artemia* in wells of a 6-well plate with 8 mL MEM (diameter 3.48 cm wells) for 30 min (6 larvae per 6-well plate). The density of the *Artemia* is adjusted to 150 *Artemia* per ml. Then, larvae were anesthetized with MS-222 (Argent Chemical Laboratories, Redmond, WA, USA) and fixed with 4% paraformaldehyde (PFA) (Servicebio, Wuhan, China) overnight. Photographs of the orange area of *Artemia* in the digestive tract were taken by a stereomicroscope and measured with Image J1 software. The amount of food ingested by medaka larvae was developed by the procedure described previously [66]. 

### 4.4. Growth Performance and Survival Rate

For growth performance and survival rate, 6 dph larvae were fed with abundant *Artemia* twice daily for 10 days. Twenty WT, *sws2a*^−/−^, and *sws2b*^−/−^ medaka larvae were randomly selected, anesthetized, and fixed with 4% PFA for total length measurement. The experiment was repeated 3 times.

### 4.5. Behavioral Tests

Behavioral tests were conducted between 15:00 and 17:00 using the DanioVision Observation Chamber (Noldus Information Technology, Wageningen, The Netherlands) linked to the EthoVision XT13 software. The 6 dph larvae were plated onto 24-well plates (diameter 15.6 mm wells) with 1 mL MEM (individual larvae per 24-well plate). Further analysis was performed using custom Open Office Org 2.4 software.

Light response: The larvae were acclimated for 30 min in the dark at 28 °C and then tracked the movement of larvae for 4 min, with 2 min of the dark period and 10 s of the light stimulation period [46]. The average swimming speed (cm/s) for dark and light were collected every 2 min and 10 s, respectively.

Light/dark behavior analysis: The larvae were acclimated for 10 min at 28 °C, and the larval locomotor activity was tested in response to dark–light conversion (3 min light/3 min dark/3 min light/3 min dark) based on the protocol by Huang et al. [67], with modifications to the transition stimulation time. The average swimming speed (cm/s) for each individual larva was collected every 60 s.

### 4.6. Hematoxylin-Eosin (H&E) Staining

Medaka at 6 dph were preserved in 4% PFA for 24 h and dehydrated in 70–100% ethanol, embedded in paraffin, and sectioned at thickness of 4 μm (Leica, Heidelberg, Germany). Then, the sections were stained with hematoxylin-eosin (H&E) according to standard protocols. The slides were imaged by slice digital scanning (Pannoramic250, Pannoramic250 MIDI, 3D HISTECH).

### 4.7. RNA Isolation and Quantitative RT-PCR

All fish were sampled in the light phase of the light/dark cycle. The adult fish tissues (n = 3) and the two larval eyes of the 6 dph medaka (n = 6) were collected and frozen in liquid nitrogen. An equal amount of RNA was extracted from each sample according to the RNAiso instruction, and the cDNA was reversed transcribed by the HiScript^®^ III 1st Strand cDNA Synthesis Kit (Vazyme, Nanjing, China). The reaction system (20 μL) contained 1 μL cDNA template, 10 μL SYBR (Vazyme, China), 0.4 μL of each primer, and 8.2 μL ddH_2_O. The cycling parameters were 95 °C for 30 s, 40 cycles at 95 °C for 10 s, 58 °C for 30 s, and melting curve from 65 °C to 95 °C (gradually increasing 0.5 °C s^−1^), with data acquired every 6 s. The results were normalized to *β-actin,* and relative transcript abundances of genes were performed using the 2^−ΔΔCt^ value method [68]. All primers are shown in Appendix A.

### 4.8. Statistical Analysis

All results are presented as means ± S.E.M (standard error of the mean), and the normality of the data was first tested by the Shapiro–Wilk test. The differences among three groups were analyzed by one-way ANOVA and Duncan’s multiple-range test, and (*p* < 0.05) was considered a significant difference. The differences between the two groups were determined with Student’s *t*-test, and statistical significance was determined at *p* < 0.05.

## Figures and Tables

**Figure 1 ijms-24-08786-f001:**
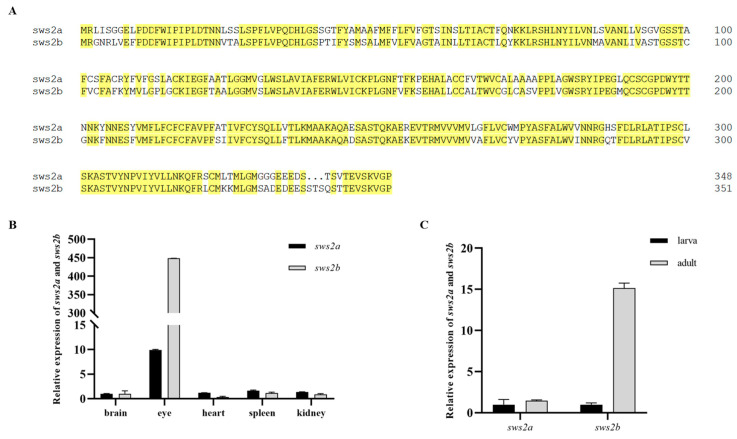
Conservation analysis and expression characterization of *sws2a* and *sws2b* in medaka. (**A**) Sequence alignment of amino acids in *sws2a* and *sws2b* (n = 3). (**B**) Differences of *sws2a* and *sws2b* mRNA expression in different tissues of medaka adult. (**C**) Expression of *sws2a* and *sws2b* in larva and adults (n = 6).

**Figure 2 ijms-24-08786-f002:**
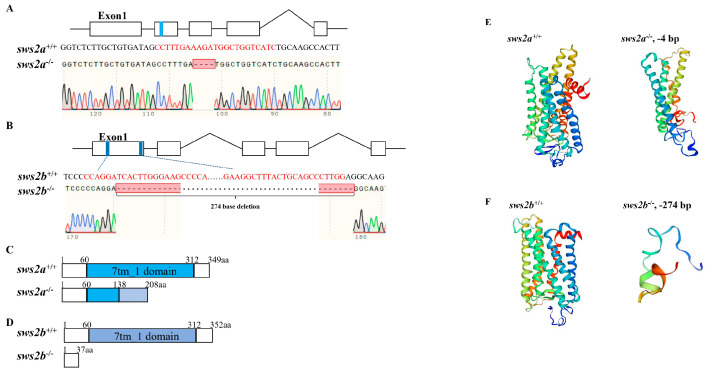
Generation and characterization of *sws2a* and *sws2b* mutant medaka. (**A**,**B**) Design target sites of *sws2a* and *sws2b* based on CRISPR/Cas9 technology. Exons are represented by boxes, and single-guide RNAs (sgRNAs) are labeled with blue color in exon. The sgRNA sequences are highlighted in red, and the −4 bp and −274 bp deletions are indicated by sequencing validation. (**C**,**D**) Illustration of deduced protein structure of WT and *sws2a* and *sws2b* mutants. These numbers represent amino acid positions from the initiation codon. (**E**,**F**) The *sws2a* and *sws2b* protein tertiary structure prediction of WT and *sws2a* and *sws2b* mutants.

**Figure 3 ijms-24-08786-f003:**
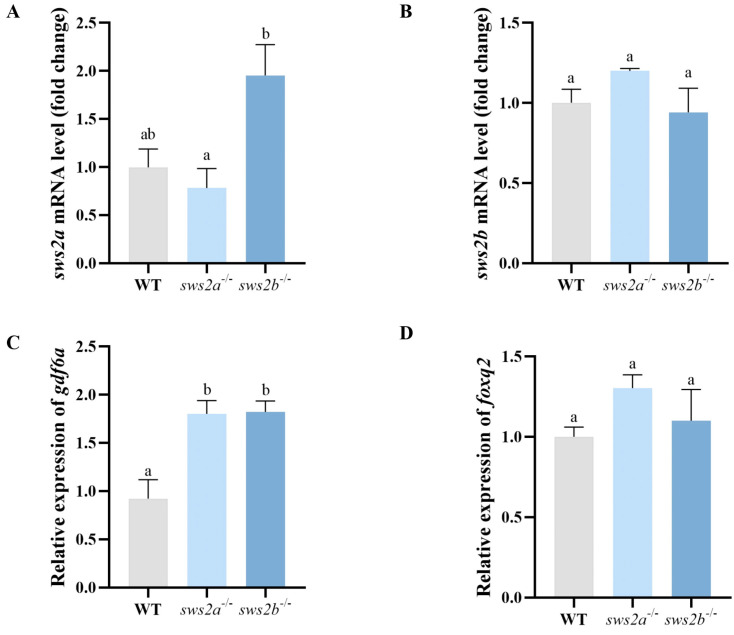
The mRNA levels of *sws2a* and *sws2b,* and the transcription factors *gdf6a* and *foxq2* in the larval eyes at 6 dph. (**A**) *sws2a*. (**B**) *sws2b*. (**C**) *gdf6a*. (**D**) *foxq2*. All data are expressed as the mean ± SEM (n = 6). Vertical bars not sharing the same letter are significantly different (*p* < 0.05).

**Figure 4 ijms-24-08786-f004:**
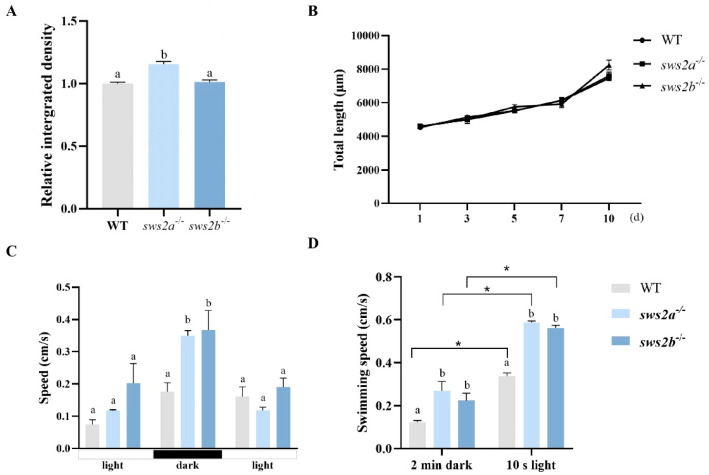
Analysis of feeding and the behavioral tests among WT, *sws2a*^−/−^, and *sws2b*^−/−^ medaka at the larval stage. (**A**) Relative levels of ingested *Artemia* shown in the digestive tract (n = 60). (**B**) Total length of medaka from 1 to 10 days from first feeding (n = 15/group). (**C**) The statistical analysis of swimming speed during photoperiod stimulation period in WT, *sws2a*^−/−^, and *sws2b*^−/−^ larvae. (**D**) The swimming speed during the last 2 min dark and the first 10 s of the 2 min light period (* *p* < 0.05 by Student’s *t*-test). Vertical bars not sharing the same letter are significantly different (*p* < 0.05).

**Figure 5 ijms-24-08786-f005:**
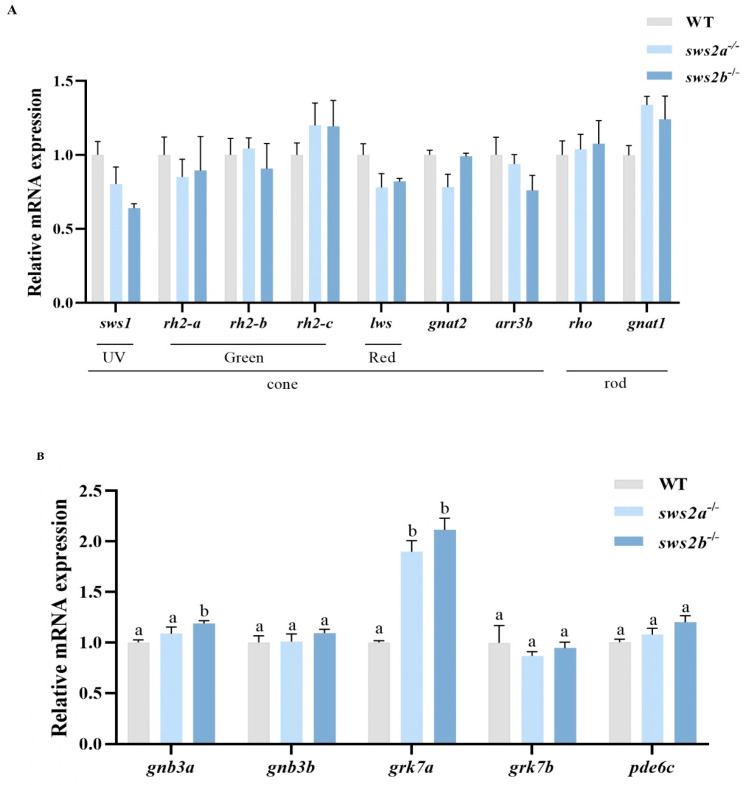
The transcriptional levels of opsin- and phototransduction-related genes in the larval eyes at 6 dph. (**A**) Genes involved in opsin (*sws1*, *rh2-a*, *rh2-b*, *rh2-c*, *lws*, *gnat2*, *arr3b*, *rho*, and *gnat1*). (**B**) Genes involved in phototransduction (*gnb3a*, *gnb3b*, *grk7a*, and *grk7b*, and *pde6c*). The data are represented by mean ± SEM (n = 6). Vertical bars not sharing the same letter are significantly different (*p* < 0.05).

**Figure 6 ijms-24-08786-f006:**
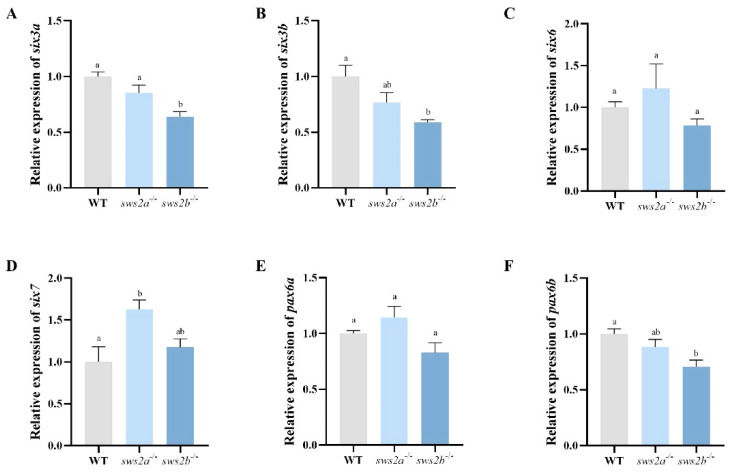
The results of the qRT-PCR analysis of eye development genes in larval eyes at 6 dph. (**A**) *six3a*. (**B**) *sin3b*. (**C**) *six6*. (**D**) *six7*. (**E**) *pax6a*. (**F**) *pax6b*. All data are expressed as the mean ± SEM (n = 6). Vertical bars not sharing the same letter are significantly different (*p* < 0.05).

## Data Availability

All data are available from the corresponding author by request.

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
