# Peer review of "Knockout of sws2a and sws2b in Medaka (Oryzias latipes) Reveals Their Roles in Regulating Vision-Guided Behavior and Eye Development"

_ijms, 2023, doi:10.3390/ijms24108786_

Round 1

Reviewer 1 Report

Ke Lu et al. Knockout of sws2a and sws2b in medaka (Oryzias latipes) re- 2 

veals their roles in regulating vision-guided behavior and eye 3 development 

This manuscript reports the CC induced ko of two medaka opsins and some studies to phenotype the consequences of this gene loss. The manuscript, though potentially interesting, suffers from poor English which makes it often difficult to follow descriptions and conclusions. The discussion is a collection of literature citations that in part contradict the authors conclusions. An in-depth discussion is missing.

Furthermore important experimental data is missing. A gene knock-out must be accompanied by a detailed description of gene expression in relevant tissue. Mere qPCR analysis to quantify expression levels is not sufficient. The choice of behavioural tests is not appropriate and requires additional tests to support the conclusions. 

This manuscript requires a major revision to achieve a publishable form.

Improve English: the manuscript should be edited by a native English speaker. In many instances the English is poor and statements misleading or difficult to understand. It is very time consuming and annoying to review a manuscript that is written in such a poor style.

Some examples:

line 30: "opsins that bind to the retina" is a wrong statement and should be corrected. Opsins are membrane proteins that are located in the ONL of the retina as well as many other cell types.

line 91 ff: Protein domain measurements 91 further elucidate the knock-out results (Figure 2E-F). What are “protein domain measurements? Do the authors mean “protein domain predictions?

line 93 ff: we performed transcript levels of sws2a and sws2b in WT, sws2a-/- and sws2b-/- medaka by qPCR. Probably the authors mean: we analysed transcript levels……. 

Other points

Genotyping of ko fish: how were heterozygous and homozygous mutant individuals identified after the Crispr/cas knock-out?

mRNA levels (fig 3 and 6): were the levels measured in retinal tissue (dissected eyes)? The authors should mention this.

Expression of sws2a/b: a detailed expression analysis by whole mount in situ analysis (WISH) is missing. In view of the phenotype and its interpretation this is important data. The retinal expression: are the genes expressed exclusively in the ONL? Are sws2a/b expressed in the brain?

The authors show a decreased expression level of several genes (fig5): was the expression level determined of retinal tissue only? Was the retina morphologically normal (retinal layers, this is not visible in fig S3). Did the authors analyse the size of the retina/number of cells/cell death? The reduced expression could be the result of fewer cells in the ONL. The authors mention that their work is in accordance to a zf lca5 ko (lines 353), there a cone-rod dystrophy has been reported. Also the authors speculate that loss of sxs2b may affect the retinal development in medaka larvae. 

The authors describe behavioural phenotypes of the single mutants. In view of possible functional redundancy/compensation of sws2a and b (discussion lines 339-341) it would be interesting to examine the double mutant.

behavioural tests:

line 476/477: what are the modifications that were introduced? The authors need to describe them to ensure that experiments can be replicated in other labs.

Is the analysis of the swim speed after a light-dark transition the appropriate test when examining behaviour of opsin ko mutants? A visual acuity test (stripe assay) would provide important information about retinal function.

The feeding assay as described does not really test visual prey capture ability since the hatchling are tested in an environment with “abundant” artemia naupli. Thus prey can be captured also by random contacts. The authors should repeat this test with a low number of prey, such that visual prey detection is limiting. This should be complemented by visual acuity tests (see above). 

lines 362ff: do the authors imply that upregulation of gdf6s in mutant retinae is a compensatory effect to the loss of sws2a/b? The speculation is not explained.

lines 367 ff: all the zf mutants mentioned show a reduced foraging behaviour which is in contrast to the increased feeding in sws2b medaka ko. The authors should explain and discuss this difference instead of jut citing literature.

The statement that the authors "found enhanced vision sensitivity in regulating vision-guided behavior due to the up-regulated genes related to phototransduction” lacks experimental support and is pure speculation. 

Reviewer 2 Report

In this paper, the authors provide a sound and comprehensive examination of the role of sws2a and sws2b in medaka retina development. The results are presented clearly, the conclusions are sound, and the discussion is adequate. However, it is this reviewer's opinion that in its current state, this work requires moderate English language editing before it can be accepted for publication. I have highlighted some changes for implementation below. Notably, this list is not comprehensive, and in addition to the items below, the authors should consider the entire manuscript when performing the suggested additional editing.

MINOR

Language editing

In general, define all acronyms (e.g., Sw2, gdf6a) upon first use in the abstract and then again in the main text of the manuscript before using the acronym consistently throughout.

Use consistent formatting of gene and protein names with conventions for fish throughout the document (e.g., Abstract, line 16: gdf6a – Protein, Gdf6a; gene, gdf6a). Note, that in cases where the gene name is the first word in a sentence, the gene name should be capitalized, but still italicized (e.g., Line 95 – “sws2a-/- and sws2b-/- medaka both had no significant decrease in sws2a and sws2b mRNA…” Sws2a-/- and sws2b-/- medaka both had no significant decrease…”)

Please review the use larvae (plural noun) and larval (adjective) throughout the manuscript and make sure that the terms are used correctly (e.g., Line 123: “We examined the mRNA levels of phototransduction - related genes in larval of the WT” – should be larvae).

Please ensure the use of plurals where necessary (e.g., Line 88-90 – “Compared with WT

medaka, sws2a and sws2b homozygous mutants showed 4 bp and 274 bp deletion, respectively, and the deletion resulted in the early termination of translation of the entire seven-transmembrane domain (7tm_1) in SWS2 (Figure 2A-D).”  Should be “deletionsthese deletions.” Please review and revise the use of plurals here and throughout the entire manuscript.

Please check for the use of consistent spacing conventions throughout the manuscript (e.g., “6dph” vs. “6 dph” or “…maxima[51].” vs. “…maxima [51].” or “10h” vs. “10 h”).

Please correct the use of SimSun font throughout the manuscript (e.g., “°C” vs. “°C”).

Please consistently use WT or wild-type throughout the manuscript.

Complete a spell-check after all edits have been made.

More specific examples:

Abstract

Line 17 -  “sws2a-/- and sws2b-/- mutants larvae sws2a-/- and sws2b-/- mutant larvae

Line 19 - “The enhanced of vision-guided behavior The enhanced vision-guided behavior

Line 20 - “the result of up-regulated of phototransduction genes the result of up-regulated phototransduction genes

Line 22 - “sws2a and sws2b knockout increase sws2a and sws2b knockouts increase…

Lines 20-21 and Introduction 70-71 - “Additionally, we also found that sws2b effects the expression of eye development genes, while sws2a is unaffected.”

1 “effects affects

2. Sws2a is unaffected? Do you mean that unlike sws2b, sws2a does not affect the expression of eye development genes? Then the sentence should read “Additionally, we also found that sws2b affects the expression of eye development genes, while swa2a does not”? Please revise accordingly.

Results:

Line 93-94: “we performed transcript levels of sws2a and sws2b in WT, sws2a-/- and sws2b-/- medaka by qPCR.” Do you mean “we examined the transcript levels of…”? Please revise for clarity.

Line 100: “which showed 4 bp and 274 bp are deletion in sws2a and sws2b mRNA levels in sws2a-/- and sws2b-/- mutants (Figure S2)” “which showed 4 bp and 274 bp deletions in sws2a and sws2b mRNA levels…”

Please review and revise for the correct use of comparisons throughout the manuscript. (e.g., Line 110: “The food intake of Artemia in sws2a-/- medaka larvae was significantly higher than WT and sws2b-/- “…higher than that of WT and…”

Discussion:

Line 332 – “In contrast to mammals that have lost sws2 and rh2 genes through evolution’s choice”. Please avoid the use of the possessive “ (‘s)” in academic writing. Also, the phrase “evolution’s choice” is inappropriate here. Do you mean “natural selection”? Or do you mean “gene deletion because of the common evolutionary process? Please revise to improve the English language use here.  

Lines 357-359 – Disruption of foxq2 in zebrafish showed the loss of sws2 cone in 5dpf larval and further results indicated that foxq2 was an activator of sws2 transcription and inhibited sws1 expression.Do you meansws2 cone expression”? Please revise accordingly for clarity, and correct the word choice (i.e., larval should be larvae).

Figure legends:

Figure 3. “The mRNA levels of sws2a, sws2b and the transcription factors.” Do you mean “…and the transcription factors gdf6a and foxq2”? Please revise to complete the sentence for clarity.

Figure 4. “Analysis of feeding and the behavioral tests among WT, sws2a-/- and sws2b-/- at the larval 241 stage.” Do you mean “…among WT, sws2a-/- and sws2b-/- medaka at the larval stage”? Please revise for clarity.

Figure 5. “The transcriptional levels of opsin and phototransduction related - genes at 6dph” “… phototransduction-related genes at 6 dph.” AND “(B) genes involved” “(B) Genes involved”

Figure 6. “The qRT-PCR results of eye development genes…” Do you mean “The results of the qRT-PCR analysis of eye development genes…” Please revise to improve English language fluency and use. AND “(G) .All data are expressed as the 326 mean ± SEM (n = 6).(G) All data are expressed...”

Line 396: “Some recently studied cartilaginous fishes also have not detected the expression of sws2 “… also show no expression of sws2

Materials and Methods:

Please use complete product citations, including the manufacturer’s name and the city, state, and country of the company headquarters (i.e.,  ThermoFisher Scientific, Waltham, MA USA)

Round 2

Reviewer 1 Report

The authors have addressed most of the issues raised in an appropriate way and the revised manuscript is now publishable.